# Cross-Sectional Analysis of University Students’ Health Using a Digitised Health Survey

**DOI:** 10.3390/ijerph17093009

**Published:** 2020-04-26

**Authors:** Pier A. Spinazze, Marise J. Kasteleyn, Jiska J. Aardoom, Josip Car, Niels H. Chavannes

**Affiliations:** 1Department of Primary Care and Public Health, Global Digital Health Unit, School of Public Health, Imperial College London, London W6 8RP, UK; josip.car@ntu.edu.sg; 2Centre for Population Health Sciences (CePHaS), Lee Kong Chian School of Medicine, Nanyang Technological University, Singapore 308232, Singapore; 3Department of Public Health and Primary Care, Leiden University Medical Center, 2333 ZA Leiden, The Netherlands; M.J.Kasteleyn@lumc.nl (M.J.K.); J.J.Aardoom@lumc.nl (J.J.A.); n.h.chavannes@lumc.nl (N.H.C.); 4National eHealth Living Lab, Leiden University Medical Center, 2333 ZA Leiden, The Netherlands

**Keywords:** health screening, health survey, university student health, young Asian adults

## Abstract

University student years are a particularly influential period, during which time students may adopt negative behaviours that set the precedent for health outcomes in later years. This study utilised a newly digitised health survey implemented during health screening at a university in Singapore to capture student health data. The aim of this study was to analyze the health status of this Asian university student population. A total of 535 students were included in the cohort, and a cross-sectional analysis of student health was completed. Areas of concern were highlighted in student’s body weight, visual acuity, and binge drinking. A large proportion of students were underweight (body mass index (BMI) < 18.5)—18.9% of females and 10.6% of males—and 7% of males were obese (BMI > 30). Although the overall prevalence of alcohol use was low in this study population, 9% of females and 8% of males who consumed alcohol had hazardous drinking habits. Around 16% of these students (male and female combined) typically drank 3–4 alcoholic drinks each occasion. The prevalence of mental health conditions reported was very low (<1%). This study evaluated the results from a digitised health survey implemented into student health screening to capture a comprehensive health history. The results reveal potential student health concerns and offer the opportunity to provide more targeted student health campaigns to address these.

## 1. Introduction

Health screening has been adopted by several countries, particularly in Asia and the United States, as a mandatory requirement for entry into tertiary education institutions. Screening provides an opportune moment to capture baseline health information for the evaluation of student health during their time at university. Health screening has traditionally been conducted using paper and pen assessment forms. By implementing a digitised health survey with algorithm-based adaptive questionnaires, we can individually tailor questionnaires, improve data capture and efficiency, and concurrently obtain valuable population health insights to deliver targeted health interventions.

### 1.1. Student Health Screening

Universities utilise health screening to determine the minimum physical health requirement for incoming students, and to ensure the safety and wellness of the school population. Universities are particularly at risk for communicable disease outbreaks, because of the diversity of geographical origins, high degree of person-to-person interaction, and relatively crowded dormitory settings [1]. Requirements for entry into the university or designated courses vary by institution and can, for example, include proof of vaccination or other medical investigations, e.g., blood test or chest X-ray (CXR).

University student years are an important period in emerging adulthood, representing a pivotal transition from both secondary to tertiary level and beyond graduation. During this time, individuals generally experience greater autonomy, with increased responsibility and life stressors that have significant implications for their mental health, wellbeing and engagement. This period also sets the precedent for health behaviours into adulthood.

Perhaps attributable to increased autonomy, a significant proportion of the student population begin engaging in risky health behaviours, including increased tobacco, alcohol, and substance use; unprotected sexual intercourse; and decreased health-protective behaviours, including physical activity and healthy eating [2,3,4]. Psychological distress amongst students in higher education has been widely studied and is a global health concern [5,6]. Causes for distress can include academic and financial pressure associated with an increased workload, pressure to perform, student loans, leaving home, and the establishment of new social networks [7]. Although the health impact of these factors may vary according to different social and cultural contexts, university students are at increased risk of depression, anxiety, burnout, and suicide [5]. In Singapore, the total lifetime prevalence of any mental disorder surpassed 20% in 2016, with the highest risk of onset of depression being in those aged 18 to 34 years old [8]. Thus, with the global number of students enrolled in higher education forecasted to more than double to 262 million by 2025 [9], the increased demand on student support services is becoming more apparent. Resources will need to be adapted adequately and appropriately in order to cope with the demand.

When the health and wellbeing of students are compromised, this can also negatively affect several functional areas of daily activity and quality of life. Academic achievement and the poor health status of students is inversely correlated. Health-related issues, such as asthma, malnutrition, obesity, chronic stress, and risk-taking behaviours, such as aggression and violence, drunk driving, unsafe sexual activity, unhealthy eating, binge drinking, physical inactivity, and substance use, are all associated with low academic performance [10].

### 1.2. Singapore Healthy Campus Initiative

Globally, there are several campus-related initiatives aimed at improving the overall health of college students, including modifications to campus infrastructure, services (i.e., counseling and support groups), and programs (e.g., Stanford’s Resilience Project, featuring academic skills coaching) [11]. The Ministry of Health, Singapore, launched one such initiative in 2018, with several aims, including improving health during the period of entering versus leaving university, training graduates to become lifelong health ambassadors in society, and creating a campus environment that proactively promotes healthy lifestyles.

As part of this initiative, the aim of this study was to evaluate the baseline health status of students entering university in Singapore (from here on referred to as baseline health information), creating a database for future comparison.

## 2. Materials and Methods

### 2.1. Study Population

A cross-sectional study was conducted using student health data collected through a newly implemented digitised health survey. The data were collected over six days at one of Singapore’s national university on-campus health centres during the 2019 matriculation health screen.

### 2.2. Digital Health Survey

The digital health survey was developed based on the current matriculation health survey form, including the following details: personal particulars (faculty, course, residence, residential status), demographics, medical history, medication and allergy history, family history (chronic inheritable conditions, i.e., hypertension, cancer etc.), tobacco and alcohol consumption, and mental health/psychiatric history. Minor edits were made to each section in terms of greater granularity of questions, with the addition of screening questions for alcohol misuse, tuberculosis symptoms, and caregiver status (see Appendix A). A validated questionnaire was used for the assessment of alcohol misuse: the Alcohol Use Disorders Identification Test–Concise (AUDIT-C). This is a brief alcohol-screening instrument that reliably identifies persons who are hazardous drinkers or have active alcohol use disorders (including alcohol abuse or dependence). The AUDIT-C is a modified version of the 10-question AUDIT instrument that gives a score on a scale from 1 to 8. A score of ≥3 and ≥4 for females and males, respectively, is considered positive for identifying hazardous drinking or active alcohol use disorders.

The questionnaire was developed and adapted in consultation with the Health Promotion Board (HPB), Ministry of Health (MOH), and university interdepartmental support. Subsequently, it was tested on student and research staff for understanding and time for completion. The questionnaire was also translated into Chinese by means of a forward–backwards translation method, to allow Chinese international students/staff to complete it with greater ease and understanding.

### 2.3. Study Procedure

Students were notified of the health screening requirements upon receiving their acceptance to the university. The majority of student health screening occurs on campus through a dedicated private clinic managed by a third-party healthcare provider. A small minority of students (<10%) complete their health screen through another private healthcare facility in Singapore, or for international students, in their country of origin prior to arriving in Singapore. These data were captured through the original paper matriculation health check form, and were not included in the study cohort.

Students who attended the on-campus health clinic for screening were required to complete a digital health survey, in addition to the current screening process. As the healthcare provider still needed to maintain a paper record for their files, the digitised records were in addition to current processes and not a replacement.

The current matriculation health screen is comprised of six steps (see Figure 1):1.Registration: students register at the clinic’s reception desk with their valid identification and matriculation health check forms. The staff enter the students’ details into the system and print the patient stickers. These are attached to the forms as well as a blood lab form. The forms are stamped with the date and clinic details and stapled together. A slip with the student’s queue number is printed and attached to the forms with a paper clip. This process takes 1–2 min—however, with only 1–2 staff completing this, students are required to wait during peak times.2.Measurements: height, weight, body mass index (BMI), and visual acuity are assessed at a single station. Height and weight are assessed using an automatic standing scale, from which BMI is calculated. The eye test is done using a Rosenbaum chart (numbers chart) at 6 m distance, noting whether the participant is using corrective lenses or not. The results are hand-recorded on the student forms, and no further action is taken. There is one healthcare worker at this station, and the station takes 2–3 min to complete. Total time at the station, including waiting time, can range from between 2 (with no queue) to 10 min during busy times.3.Urine test: students are given a bag/cup with a urine dipstick, sent to the bathroom to cover the stick in urine, and asked to return to show the results to the nurse. The test is assessed by a nurse. Abnormal results, including any trace elements of glucose, blood, or protein are generally managed following a protocol (see Figure 2). There is one healthcare worker at this station. On average students, spend 10 min at this station from the time of collection of dipsticks to completion. The staff confirm students’ queue numbers or names, and manually write the results into their form. Female students who are currently having their menstrual period need to return 7 days after the end of their period to complete the urine test.4.Blood tests: specific blood tests are required for international students and for students entering specific courses:
Hepatitis B—medicine and biomedical science (traditional Chinese medicine), as well as international students;Hepatitis C—medicine;HIV (human immunodeficiency virus)—medicine and international students.

Abnormal blood results are followed up by the attending doctor for further investigation. There are two phlebotomists manning this station, and the time to complete blood tests is on average 2 min per student.

5.Chest X-ray (CXR): all students require a CXR to assess for evidence of pulmonary tuberculosis, as well as other chest pathologies, including current chest infections or structural abnormalities. There are two to three staff members at this station, one registering, one coordinating, and the other one conducting the X-rays. On average, students spend around 4–25 min at this station. The X-ray itself only takes 2–3 min; however, the preparation and waiting time is required, as female students are requested to remove their bra and tops and wear a gown, as well as put their hair in a bun to avoid artefacts on the X-ray.6.Doctor consultation: students are required to complete a doctor’s consultation, where their blood pressure (BP), pulse, colour vision, and a physical examination are completed. There are generally four doctors on duty, with one designated to normal health consultations and the rest focused on matriculation health screening. After 5:30 pm, there is only one doctor to attend to the remaining students still to be seen. The consultation times range from 2–10 min, with an average of 3–4 min per student which varies from doctor to doctor.

The stations may be attended in any order following registration; however, the first few students that register are generally seen first by the doctors, due to availability. As the queue builds up, students complete the other stations prior to being seen by a doctor. The students are allocated to a consultation room according to their queue number, which appears on screens in the waiting area. On completion of all stations, the students return in five working days to collect their results from the medical records office (MRO). The digital health survey was set up as an additional station that students attended at any time during their health screen.

### 2.4. Data Management

REDCap was the platform used to conduct the health survey and collect responses. REDCap is a secure, web-based research application for building and managing online surveys and databases, developed by Vanderbilt University [12]. It meets health data privacy and security requirements, including the Health Insurance Portability and Accountability Act (HIPAA) compliance. As part of the health screen, REDCap was incorporated into and hosted on the healthcare provider’s network, behind a secure firewall to prevent possible data breaches. The data collected were owned, stored, and managed by the healthcare provider, who provided the research team with access to anonymized student health data for population health research purposes. All personally identifiable information was removed from the dataset shared through the REDCap platform, including date of birth, names, addresses, and other personal information, and stored on local secure servers for analysis.

### 2.5. Equipment and Manpower

The health survey was delivered through iPads that were distributed to students in a self-contained room, with a capacity of up to 20 students at a time. Students were required to check-in by completing a paper slip including name, national identification number, and mobile telephone number. This slip was stored in a numbered container tagged to the iPad distributed to them, and returned to the student once the iPad was returned. A staff member oversaw this process to ensure that all iPads were accounted for and functional. There were a total of 50 iPads, with 25 iPads operating at any given time; the remaining iPads were left for charging when not in use, to ensure continued operation of the survey. Three iPads were assigned to staff to assist in data collection at the measurement and urine station.

### 2.6. Ethical Considerations

Consent was integrated through the current Personal Data Protection Act 2012 (PDPA) consent form that healthcare providers require all students attending the clinic to complete. An additional clause was incorporated into this PDPA to describe the use of their data for population health research (see Appendix B). In addition to this, students could opt out of sharing their data with the university for health research purposes. This study was approved by the University’s Institutional Review Board (IRB-2019-08-031, approval date: 16 December 2019).

## 3. Results

There were 758 students who registered to complete the health survey. A total of 223 students were excluded, leaving a total of 535 students included in the study cohort for analysis (see Figure 3). Sixty-three percent of participants completed the English survey, and the remainder completed it in Chinese. Those that completed the survey in Chinese were foreign students, with the exception of four Singaporean permanent residents.

### 3.1. Demographics

The largest proportion of students were undergraduates (87.0%), compared with postgraduates and males (54.0%). With regards to ethnicity, the majority of students were Chinese (75.0%), followed by Indian (12.0%), Malay (1.0%), and White European (1.0%) (see Table 1). Students in the “Other” category were of Vietnamese (*n* = 21), Indonesian (*n* = 17), Korean (*n* = 12), and Burmese (*n* = 5) ethnicity.

### 3.2. Physical Measures

There were 519 students who had their physical measurements recorded. According to the World Health Organisation (WHO) international BMI guidelines [13], 19.0% of females and 10.6% of males were underweight, whereas 12.6% of females and 31.9% of males were overweight. Three percent of females and 7.0% of males were classified as obese (BMI > 30). If we were to consider differences in BMI amongst the Asian population, where the recommended upper limit of a normal BMI is 23, this would place 25.0% of females and 50.0% of males as overweight (Table 2).

### 3.3. Medical History

A large proportion of students reported eye-related conditions (18.0%), followed by skin conditions (6.0%) (Table 3). Regarding infectious diseases, three students were Hepatitis B carriers, and one had previously successfully completed treatment for tuberculosis. Only one student reported a mental health condition; however, three students reported suffering from anxiety, obsessive-compulsive disorder, and insomnia, respectively. Concerning women’s health conditions, common conditions reported included irregular menses and severe menstrual pain.

### 3.4. Allergies and Medications

Specific drug allergies were recorded by 7.1% of students, of which 55.3% were due to penicillin or its derivatives, e.g., co-amoxiclav. There were 10.8% of students who had other, non-drug related allergies; the majority were attributable to seafood/shellfish (Table 3). The most common prescription medication used by students was for acne (isotretinoin), followed by antibiotics for current infections; however, overall, this number was less than 1%. Two percent of students used some form of traditional Chinese medication; of the reasons specified, menses was one of the most frequent reasons stated.

### 3.5. Family History

Hypertension, diabetes, heart disease, and cancer were reported most frequently in students’ families (see Table 3). Mental health-related conditions in family members, including depression, bipolar disorder, anxiety, schizophrenia, and alcohol abuse was reported in up to 2.8% of students surveyed, with the majority of these related to depression and anxiety.

### 3.6. Smoking

The prevalence of smoking amongst students surveyed was low, with eight students reported to currently smoke; the ratio of male to female smokers was 7:1 (see Table 4). Six students had started smoking between 19–22 years of age, and two students had started smoking younger than 16 years. These students were also smoking an average of 4–5 cigarettes a day. Three had attempted to quit smoking using different methods, including nicotine replacement, smokeless tobacco, and abrupt cessation (“cold turkey”); however, none were successful. Only two of these smokers indicated that they would like to receive help or advice regarding smoking cessation.

### 3.7. Alcohol

The majority of students had not consumed alcohol (64%). Among those that did, 61% were male, and the prevalence of alcohol consumption in the postgraduate and undergraduate population was comparable. Analysis of health survey data collected showed that nine of the females and eight of males who consumed alcohol had hazardous drinking habits, with 15.9 of these students typically drinking 3–4 alcoholic drinks each time (see Table 5).

## 4. Discussion

The digitised health survey was implemented in the final week of matriculation health screening for the 2019/2020 cohort, and was subjected to various operational challenges, including acquiring approval from the healthcare centre, limited manpower, and increased demand on the centre during that time. Therefore, the data captured constituted 9% of the total number of matriculating students (*n* = 8152 full-time undergraduate and postgraduate students). Despite these challenges, the student sample was representative of the overall intake in terms of gender proportions (total intake: 51.7% male, 48.3% female vs sample intake: 54% male, 46% female); however, the postgraduate students were underrepresented (total intake = 22%, sample group = 13%). On average, the postgraduate students represent a slightly older age group at the university, and hence a younger population may have been overrepresented. Due to the operational design of the matriculation health screen, students are segregated into groups, beginning with national servicemen (NSF), local Singaporean undergraduates and postgraduates, and ending with international students. As the data were captured in the last week of screening, most of the data were from international students, and thus were not representative of the matriculation intake or the Singapore population, in terms of ethnicity. The international student cohort underrepresented the Malay and Chinese ethnicities and over-represented the Indian and “Other” ethnicities [14].

Physical measurements, including height and weight, were collected using a digital scale, and BMI was auto-calculated. BMI, as a measure of obesity, has poor sensitivity and specificity, and does not take into account the influence of age, sex, bone structure, fat distribution, or muscle mass. Although BMI shows a correlation with body fat percentage (BF), it fails to discriminate between BF and lean mass. A more accurate measure of BF could be acquired through bioelectrical impedance analysis or waist circumference. Waist circumference is highly correlated with intra-abdominal fat mass, estimated by ultrasonography and MRI [15]. Waist circumference and waist-to-hip ratio as indicators of abdominal obesity have been shown to be better predictors of the risk of future disease than BMI [16]. Both these measures have also shown associations with diabetes mellitus, metabolic syndrome, cardiovascular disease, and certain cancers [17,18,19].

A higher BMI may indicate a higher BF or higher lean/muscle mass. Half of the males in our sample population who were classified as overweight (using the WHO international guidelines) may thus include those with high lean body mass. The assumption is that the majority of males returning from compulsory national service would have a higher lean body mass as a result of physical training.

Although a high BMI in males may be due to higher lean/muscle mass, we cannot rule out the prevalence of actual obesity amongst the student cohort. Currently, an estimated 8.7% of Singaporeans are classified as obese (BMI ≥ 30) [20]. Singapore has undergone rapid changes in diet, including increased intake of calories, sugar, and salt, with less vegetables and fruit. Young Singaporean adults (18–29 years) consume the most rice or porridge dishes, fast food, and soft drinks compared with other age groups [21]. Such dietary changes are likely to contribute not only to the increasing prevalence of diabetes, but also to the risk of cardiovascular events [22]. Diabetes is a global epidemic, which is particularly pronounced in Singapore, with a prevalence rate of 10.5%, higher than that of the world’s average of 8.8%.

The other concerning result is the prevalence of students being underweight (>10%). Compared to studies done with United States high school students and female university students in South Africa, where the rates were 1.5% [23] and 7.2% [24], respectively, this number is alarmingly high. Although the cause could be related to a number of factors, including metabolic disorders and physical activity, eating disorders are one area of concern. Eating disorders within the Asian population have received little attention, as they have predominantly been associated with western culture [25]; however, the prevalence of eating disorders within Asian populations is increasing [26]. In a seven-year retrospective study of 213 patients with eating disorders in Singapore, 75.8% of participants were between 12 to 20 years old, of Chinese ethnicity, and female (female to male ratio = 96:4) [27]. Peer, cultural, and societal pressure provide a strong impetus to develop eating disorders, and there are several health repercussions, including amenorrhea, osteoporosis, increased risk of fractures [28], and weak immunity [29]. Further studies could be done to evaluate perceptions and risk factors for eating disorders, especially in the underweight segment.

The most commonly reported health condition was myopia (10.8%), but this may also have been greatly underreported. Myopia is one of the most prevalent eye disorders globally [30], especially in Asia; therefore, students may have underreported this. Based on data acquired from the visual acuity assessment, it was noted that 68% of students underwent the test using corrected vision, i.e., either with glasses or contact lenses. The reason for corrected vision was not questioned—hence, whether all of these cases were attributable to myopia or another eye-related condition is indeterminable. However, considering that the prevalence rates in Southeast Asian countries are generally higher than in other parts of the world, it can be assumed that myopia would account for the majority of these cases. The prevalence of myopia is increasing, and has become an important public health issue in Singapore. Singapore has one of the highest prevalence rates of myopia in the world. The prevalence of myopia in seven- to nine-year-olds in Singapore ranges from 29.0% to 53.1%, respectively [31], and exceed 70% upon completion of college [32]. The exact pathogenic mechanisms of myopia are still not clear, although evidence suggests that it is likely due to hereditary and environmental factors [33]. Singapore population-based prevalence studies have indicated an increased prevalence of myopia in Singaporeans with higher levels of education, better housing, higher individual monthly income, and occupations associated with near work (including reading, writing, computer use, and playing video games) [34]. This may account for a particularly high prevalence amongst Singaporean university students. The age of onset of myopia is commonly between 5 and 15 years of age [35]; however myopia continues to progress into the early 20s. Earlier ages of onset result in a longer period of progression, which ultimately may mean more highly myopic students in the future. The risks of development of high myopia and associated complications, such as retinal tears or myopic macular degeneration in adulthood, will be considerably higher in this young cohort [36]. Further research into contributing factors, including curriculum structure and screen time (mobile, computer or TV) are warranted. A systematic review has shown that increased time outdoors is effective in preventing the onset of myopia, as well as in slowing the myopic shift in refractive error [37]. This may suggest an evaluation of teaching strategies and incorporating mandatory outdoor activities into curricula.

The other group of common conditions were related to atopic diseases, including asthma, allergic rhinitis, eczema (atopic dermatitis), and urticaria. These conditions are known to coexist and are common, especially in westernized and industrialized countries [38]. Evidence suggests that this may be due to the “Hygiene Hypothesis,” wherein allergic diseases is an unintended consequence of reduction in microbial exposure or colonization, rather than microbial infection in early life. A study in Singapore schoolchildren indicated a significant number of coexistent atopic diseases, which are also rising in prevalence [39]. Specifically, the percentage of eczema in Singapore is relatively high 13.1%, with a higher prevalence in children (20.6%) than in adults (11.1%) [40]. Eczema is associated with a lower overall health rating and life satisfaction, as well as an impaired quality of life (QoL) related to mental health. Eczema has been shown to have a worse QoL than a number of other chronic illnesses, including heart disease, diabetes, and high blood pressure [41]. Future studies could be undertaken to evaluate the severity of symptoms and relationship with academic performance and mental wellbeing.

Acne was commonly reported, and accounted for the most utilised prescription medication by students (isotretinoin). Acne is known to have a significant impact on emotions, daily activities, social activities, study/work, and interpersonal relationships. Moreover, anxiety and depression are found to be more prevalent among acne patients, including suicidal ideation [42]. Thus, in a campus setting where there is high importance placed by students on social status and relationships, acne can have a significant burden on quality of life. We could look into supportive services to help students deal with social stressors.

Mental health disorders did not appear to be very prevalent, as only one student reported a mental health disorder (depression). There were students that reported that they suffered from anxiety, obsessive-compulsive disorder, and insomnia, respectively, but did not associate this with mental health. In a Singapore Mental Health Study (SMHS) conducted in 2016, the lifetime prevalence of at least one mood, anxiety, or alcohol use disorder was 13.9% in the adult population. Depression had the highest lifetime prevalence (6.3%), followed by alcohol abuse (4.1%) [8]. These rates were significantly higher than the previous survey in 2010, which suggests either an increase in incidence of mental health disorders or an increase in awareness and willingness to report. In a society where mental health is highly stigmatized and largely uncovered by health insurance, and where attempted suicide to date is still criminalized, the number of students that declared mental health conditions may be greatly underreported. With increasing efforts on campus to create greater awareness and provide supportive services to both staff and students, future surveys could also place greater emphasis on privacy, confidentiality, and the purpose of assessment, which is to ultimately provide support and preventative health interventions.

The prevalence of smoking and alcohol use was relatively low amongst our sample cohort. Only 1.5% of students reported that they currently smoke, with an average age of onset at 18–19 years of age, which suggests that students started smoking just before or during the start of their university years. The majority of these students were male, which in the Singaporean context, may suggest that these students started smoking during their national service. The primary reason for smoking by young smokers in the 18–24 age group, as highlighted in the National Health Survey 2010, was for relaxation and stress management. This may provide an opportune time for smoking cessation efforts on campus, in order to avoid prolonged smoking habits and associated comorbid disease. In addition, some students had attempted to stop smoking and were willing to seek advice. Various smoking cessation interventions could be trialed with different groups of these students, and also leverage off peer support.

The majority of the students did not consume alcohol. However, close to 10% of males and females who were consuming alcohol were engaging in risky drinking behaviours, according to the AUDIT C score. This suggests a pattern of binge drinking amongst this group of students that do drink, which is consistent with the National Health Survey results, wherein 15.5% of young adults (18–29 years old) were most likely to binge drink compared with other age groups; this trend is on the rise [20]. Binge drinking, or otherwise heavy episodic drinking, is a pattern of risky alcohol consumption commonly found in young adults. It is often measured as having consumed four or more drinks on one occasion for women, and five or more drinks on one occasion for men [43]. Binge drinking in adolescence is associated with various negative health outcomes, including fatal and non-fatal injuries, blackouts, suicide attempts, unintended pregnancy, sexually transmitted diseases, academic failure, and violence [44]. The time spent in university is when students are most susceptible to unhealthy drinking behaviour, and thus campus health promotion campaigns should be used to address these by providing awareness, support, and an opportunity to change cultural norms and behaviour.

### Limitations and Strengths

There are a number of limitations to this study, most important of which is that the cohort was not representative of the student matriculation population. The majority of the study population were international students, who form the minority of the total student intake. The other limitation was that the nature of this health survey being incorporated into the health screening process at a clinic may influence the accuracy of reporting of students. Students may associate their health status with acceptance to the university—i.e., clearing their health screen. This could result in underreporting on behavioural patterns or clinical problems that may not be seen as favourable by the university, i.e., substance use and mental health. This may also reflect the low numbers recorded in these two domains. To some extent, there may be a level of social desirability bias or reporting bias that occurs when individuals deny or underreport engaging in what they perceive as socially undesirable behaviours. As the current purpose of the health screen is to deem the student fit for university attendance, we did not proceed beyond these requirements to capture other variables, such as behavioural health measures. However, there is potential for these measures to be incorporated into future health surveys, in order to provide a more holistic view of student health. The strength of the study was that it showed the value of a digitised health survey in terms of data collection, as the completion rate was 100% for all those that took the health survey.

## 5. Conclusions

A digitised health survey can capture a comprehensive population health profile to provide targeted health prevention campaigns. This study revealed several health concerns from a Singaporean university student cohort, including binge drinking, smoking, and weight control. Although low in prevalence, these could have several negative future health outcomes if not addressed.

## Figures and Tables

**Figure 1 ijerph-17-03009-f001:**
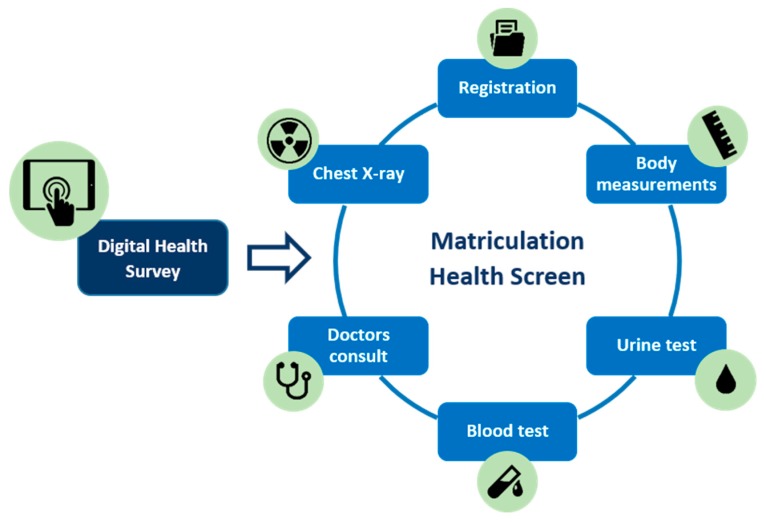
Matriculation health screen workflow.

**Figure 2 ijerph-17-03009-f002:**
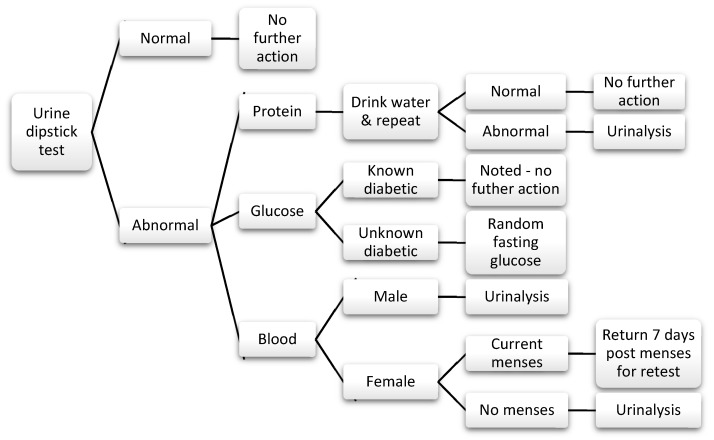
Urinalysis protocol.

**Figure 3 ijerph-17-03009-f003:**
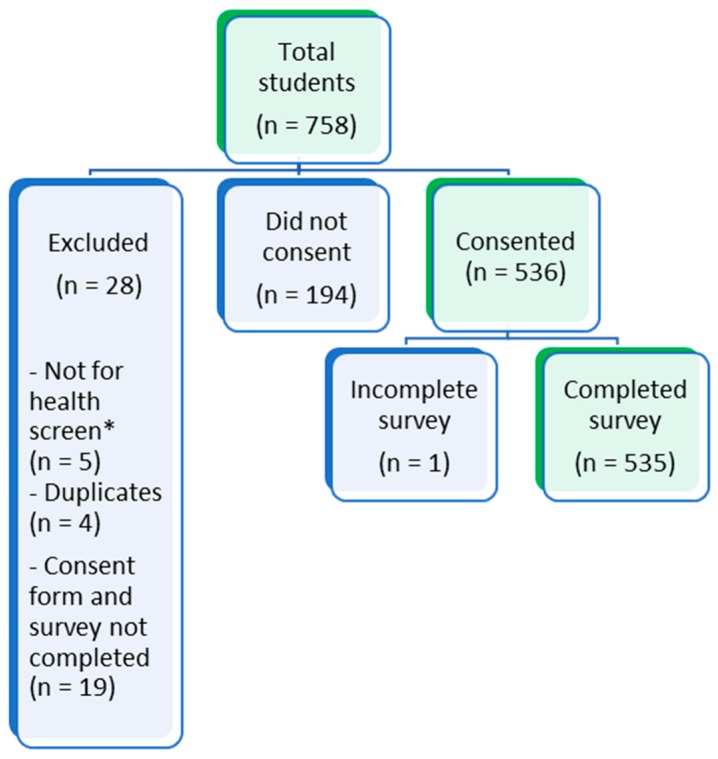
Flow diagram showing exclusion of study participants. (* students who attended clinic for other health related matters).

**Table 1 ijerph-17-03009-t001:** Total number of local and international students by ethnicity and gender.

Ethnicity	Local Students		International Students		Total *n* (%)
	Female	Male	Female	Male	
Chinese	32	21	153	193	399 (75%)
Malay	3	2	0	0	5 (1%)
Indian	1	4	29	29	59 (11%)
White European	0	0	2	1	3 (1%)
Other	1	0	29	43	73 (14%)
Total	37	27	213	266	543 *

* Note: Total number of students (543) is inflated, as this does not account for mixed ethnicity.

**Table 2 ijerph-17-03009-t002:** Total number and percentage of students, stratified according to body mass index (BMI).

Classification (WHO)	BMI	Female		Male	
		*n*	%	*n*	%
Severely underweight	<16	3	1.3%	4	1.4%
Mildly underweight	16–16.9	13	5.5%	4	1.4%
Moderately underweight	17–18.49	29	12.2%	22	7.8%
Normal range	18.5–24.9	162	68.4%	162	57.4%
Overweight	25–29.9	22	9.3%	69	24.5%
Obese	30–39.9	8	3.4%	20	7.1%
Severely obese	>40	0	0.0%	1	0.4%
Total students		237		282	

**Table 3 ijerph-17-03009-t003:** Overview of common student medical conditions, allergies, and family history of disease.

Common Medical Conditions	No. of Students	% of Total
Myopia	79	10.8%
Asthma	13	1.8%
Eczema	14	1.9%
Acne	9	1.2%
Allergic rhinitis	6	0.8%
Urticaria	5	0.7%
Hepatitis B	3	0.4%
Colour blindness	3	0.4%
Hypertension	1	0.1%
Irregular menses (females)	20	8.0 *
Severe menstrual pain (females)	15	6.0 *
**Allergies**		
Drug Allergies	38	7.1%
Non-drug allergies	58	10.8%
Alcohol	2	3.4% **
Nuts	8	13.8% **
Dust	12	20.7% **
Shellfish	12	20.7% **
Seafood	10	17.2% **
Cosmetics	4	6.9% **
**Family History**		
Hypertension	178	33.3%
Diabetes	158	29.5%
Heart Disease	76	14.2%
Cancer	66	12.3%
Hypercholesterolaemia	51	9.5%
Asthma	25	4.7%
Obesity	15	2.8%
Eczema	5	0.9%
Kidney disease	5	0.9%
Depression	5	0.9%
Anxiety	5	0.9%
Tuberculosis	3	0.6%
Dementia	3	0.6%
Schizophrenia	3	0.6%
Bipolar disorder	1	0.2%
Alcohol or substance abuse	1	0.2%
Chronic Obstructive Lung Disease	0	0.0%

* Percentage of total female students; ** Percentage of total non-drug allergies.

**Table 4 ijerph-17-03009-t004:** Demographics and attempted smoking cessation of students who smoke (*n* = 8).

Gender	Age Started Smoking	No. of Cigarettes Per Day	Attempted to Stop	Method of Cessation
Male	21	10	Yes	Nicotine replacement
Male	22	5	No	n/a
Male	15	6	Yes	Abrupt cessation
Male	19	4	No	n/a
Male	20	2	No	n/a
Male	14	3	No	n/a
Male	20	2	No	n/a
Female	19	5	Yes	Smokeless tobacco

**Table 5 ijerph-17-03009-t005:** Number of students who consumed alcohol by Alcohol Use Disorders Identification Test–Concise (AUDIT-C) score and gender.

AUDIT C	Male	Female	Total
1	73	59	132
2	19	8	27
3	16	7	23
4	6	3	9
5	1	0	1
6	2	0	2
7	0	0	0
8	1	0	1
Total	118	77	195

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
