# Peer review of "Cross-Sectional Analysis of University Students’ Health Using a Digitised Health Survey"

_ijerph, 2020, doi:10.3390/ijerph17093009_

Round 1
Reviewer 1 Report
The authors have done an excellent job in reporting the findings of a cross-sectional study using a new digital tool that could be further used as part of the health screening in the intended student population.
My only comment for the authors would be to check for grammar consistency:
e.g. "line 133: This data will be....".
The use of future tense in those lines is confusing, a past or present tense would be more appropriate.
Author Response
Dear Reviewer,
Thank you for your review and feedback. We have reviewed the paper and have corrected for all these errors.
Kind regards
Pier
- My only comment for the authors would be to check for grammar consistency: e.g. "line 133: This data will be....". The use of future tense in those lines is confusing, a past or present tense would be more appropriate.
Thank you for your feedback, we have reviewed the entire paper and have corrected for all these errors.
1) A number of corrections were made including: Line 27: drinking – drank Line 110: was – were Line 140: This – These Line 141-3: will – were, will be – were Line 145: needs – needed Line 151 – will - removed Line 204-7: will – removed Line 234: was – were Line 280: could be – were Line 320: start – started Line 328: was – were Line 337-8: this – that, constitutes – constituted Line 344: be – have been Line 349-50: represent – represented Line 362: are – were Line 388: be – have been Line 435: reporting – reported Line 443: declaring – declared Line 445: providing – provide Line 451: are – were Line 452: start – started Line 460: are – were Line 474: is – was Line 475: is – were Line 476: is – was Line 524: is – were Please note lines 148- 208 describe the current health screening process at the university and thus was not written past tense
Reviewer 2 Report
The authors of this well-structured article have as their main objective to evaluate the baseline health status of students entering university in Singapore. However it is recommended:
1) Introduction: is too long, in order to speed up the reading, a maximum of one and a half pages is patched.
2) Delete lines 244-250
3) Section 3.3. it is too long, a maximum of 5-6 lines is recommended.
4) It is recommended to reduce the size of section 3.4. to 1 maximum paragraph.
5) Limitations section. It is recommended that it be "Limitations and Strengths".
6) Describe in 3-4 lines what is the main conclusion of the study.
Author Response
Dear Reviewer,
Thank you for your time and valued feedback. We have amended the paper accordingly.
Many thanks
Pier
1) Introduction: is too long, in order to speed up the reading, a maximum of one and a half pages is patched.
2) Delete lines 244-250
3) Section 3.3. it is too long, a maximum of 5-6 lines is recommended.
4) It is recommended to reduce the size of section 3.4. to 1 maximum paragraph.
5) Limitations section. It is recommended that it be "Limitations and Strengths".
6) Describe in 3-4 lines what is the main conclusion of the study
We agree and have amended the paper accordingly.
1) The introduction has been shortened. Removed lines: 37, 43-45, 60-61, 69-70, 72-73, 91-96, 97-102
2) Removed lines 244-250 and reworded the paragraph for clarity. Line 223: “There were 758 students who registered to complete the health survey on REDCap. A total of 535 students who completed the health survey and provided consent were included in the study cohort for. analysis and 223 students were excluded (see Figure
3)We have furthermore removed all results viewable in Table 3 to avoid duplicate information.
4) Section 3.4 reduced to 1 paragraph. Removed data on supplements as well.
5) Edited to “Limitations and Strengths”. Furthermore, we have added a strength of the study, Line 484: “The strength of the study, was that it showed the value of a digitised health survey in terms of data collection, as the completion rate was 100% for all those that took the health survey.”
6) The conclusion has been reduced to 4 lines: “A digitised health survey can capture a comprehensive population health profile to provide targeted health prevention campaigns. This study revealed several health concerns from a Singaporean university student cohort including, binge drinking, smoking and weight control. Although low in prevalence, these could have several negative future health outcomes if not addressed.”

Reviewer 3 Report
Grammar: Data - plural word; thus, "are" not "is." Check Abstract. Throughout paper there is inconsistency in awareness of this. Grammatical errors throughout the paper with respect to "is" and "are." Same with "was" and "were" and "this" and "these." "Present" and "past" tenses - errors in use. Strongly suggest a careful reading by the authors. Believe they will be able to make the corrections.
Confused about number of subjects who participated in this study. Also, authors need to state the numbers of subjects for each Table.
Check "1" subject in line 432. Suggest authors recheck numbers in Tables and in text.
Author Response
Dear Reviewer,
Thank you for you time and consideration in reviewing our paper. We have reviewed the entire paper to correct for the grammatical errors highlighted and made amendments according to your feedback.
Kind regards
Pier
Reviewer 3: 1. Grammar: Data - plural word; thus, "are" not "is." Check Abstract. Throughout paper there is inconsistency in awareness of this. Grammatical errors throughout the paper with respect to "is" and "are." Same with "was" and "were" and "this" and "these." "Present" and "past" tenses - errors in use. Strongly suggest a careful reading
We have carefully reviewed the entire paper to correct for these errors.
1) A number of corrections made: Line 27: drinking – drank Line 110: was – were Line 140: This – These Line 141-3: will – were, will be – were Line 145: needs – needed Line 151 – will - removed Line 204-7: will – removed Line 234: was – were Line 280: could be – were Line 320: start – started Line 328: was – were Line 337-8: this – that, constitutes – constituted Line 344: be – have been Line 349-50: represent – represented Line 362: are – were by the authors. Believe they will be able to make the corrections.
2. Confused about number of subjects who participated in this study. Also, authors need to state the numbers of subjects for each Table.
3. Check "1" subject in line 432. Suggest authors recheck numbers in Tables and in text. Line 388: be – have been Line 435: reporting – reported Line 443: declaring – declared Line 445: providing – provide Line 451: are – were Line 452: start – started Line 460: are – were Line 474: is – was Line 475: is – were Line 476: is – was Line 524: is – were Please note lines 148- 208 describe the current health screening process at the university and thus is not written past tense.
1) Apologies for the confusion. We have reworded the paragraph in line 223: “There were 758 students who registered to complete the health survey on REDCap. A total of 223 students were excluded, leaving a total of 535 students included in the study cohort for analysis (see Figure 3).” Figure 3 was further edited for clarity including breakdown reasons for exclusion in one box and explanation for “not for health screen” as a note. Included student numbers for all tables that were not representative of total student cohort (i.e. 535) including Table 2 and 4. Table 5 total numbers were already reflected but we further clarified group as those who consumed alcohol.
2) The percentage symbol and decimal were incorrectly omitted – 1.5%. All numbers were rechecked and correlate with the tables.
Round 2
Reviewer 2 Report
No comments